AutoSCAN: automatic detection of DBSCAN parameters and efficient clustering of data in overlapping density regions

Bushra Adil Abdu 1
Kim Dongyeon 2
Kan Yejin 1
Yi Gangman gangman@dongguk.edu 1 2 3
1 Department of Multimedia Engineering, Dongguk University , Seoul , South Korea
2 Department of Artificial Intelligence, Dongguk University , Seoul , South Korea
3 Division of AI Software Convergence, Dongguk University , Seoul , South Korea
Pires Ivan Miguel
Electronic publication date: 2024 Mar 14
Publication date: 2024
Volume: 10
Electronic Location ID: e1921
Received 2023 Oct 24; Accepted 2024 Feb 12
Copyright: ©2024 Bushra et al.
Copyright year: 2024
Copyright holder: Bushra et al.
License: This is an open access article distributed under the terms of the Creative Commons Attribution License, which permits unrestricted use, distribution, reproduction and adaptation in any medium and for any purpose provided that it is properly attributed. For attribution, the original author(s), title, publication source (PeerJ Computer Science) and either DOI or URL of the article must be cited.
License URL: https://creativecommons.org/licenses/by/4.0/

Keywords: DBSCAN, Density-based clustering, Unsupervised clustering, K-nearest neighbors

Funding: The National Research Foundation of Korea (NRF) grant funded by the Korean government (MSIT) NRF-2022R1F1A1074228 Institute of Information & communications Technology Planning & Evaluation (IITP) under the Artificial Intelligence Convergence Innovation Human Resources Development IITP-2023-RS-2023-00254592 The Korean government (MSIT) and the Dongguk University Research Fund of 2023 This work was supported by the National Research Foundation of Korea (NRF) grant funded by the Korean government (MSIT) (No. NRF-2022R1F1A1074228), and was also supported by Institute of Information & communications Technology Planning & Evaluation (IITP) under the Artificial Intelligence Convergence Innovation Human Resources Development (IITP-2023-RS-2023-00254592) grant funded by the Korean government (MSIT) and the Dongguk University Research Fund of 2023. The funders had no role in study design, data collection and analysis, decision to publish, or preparation of the manuscript.

==============================
The density-based clustering method is considered a robust approach in unsupervised clustering technique due to its ability to identify outliers, form clusters of irregular shapes and automatically determine the number of clusters. These unique properties helped its pioneering algorithm, the Density-based Spatial Clustering on Applications with Noise (DBSCAN), become applicable in datasets where various number of clusters of different shapes and sizes could be detected without much interference from the user. However, the original algorithm exhibits limitations, especially towards its sensitivity on its user input parameters minPts and ɛ. Additionally, the algorithm assigned inconsistent cluster labels to data objects found in overlapping density regions of separate clusters, hence lowering its accuracy. To alleviate these specific problems and increase the clustering accuracy, we propose two methods that use the statistical data from a given dataset’s k-nearest neighbor density distribution in order to determine the optimal ɛ values. Our approach removes the burden on the users, and automatically detects the clusters of a given dataset. Furthermore, a method to identify the accurate border objects of separate clusters is proposed and implemented to solve the unpredictability of the original algorithm. Finally, in our experiments, we show that our efficient re-implementation of the original algorithm to automatically cluster datasets and improve the clustering quality of adjoining cluster members provides increase in clustering accuracy and faster running times when compared to earlier approaches.

Introduction

Clustering is the process of partitioning the data into a groups that are similar as possible given a set of data objects (Gan, Ma & Wu, 2020; Aggarwal & Reddy, 2014). For continuous data, a distance-based approach can be applied to determine the similarity between data objects. Distance functions such as Euclidean distance, (Saxena et al., 2017; Danielsson, 1980), cosine distance (Saxena et al., 2017; Nguyen, Chen & Chan, 2011) or Pearson correlation measure (Pearson, 1896; Bravais, 1844; Arabie & Hubert, 1996) can serve as a similarity measure between pairs of data objects. Closer data objects are more similar and likely to be grouped in the same class compared with objects that are farther apart. Through this mechanism, clustering algorithms are used as tools to partition a given dataset into objects classes or “clusters” that have more in common with each other than with others.

Many clustering algorithms have been developed and can be classified into several categories based on clustering strategies (Bushra & Yi, 2021). Partition-based methods operate by organizing k partitions of a given dataset, where each partition corresponds to a cluster. A partitioned group contains at least one object, and each data object belongs to only one partition. The k-means (Jain, 2010; Hartigan & Wong, 1979; Reddy & Vinzamuri, 2013) algorithm is the most representative clustering method, which assigns a partition’s centroid as the average value of the objects in the partitioned group, and performs clustering by identifying which partition’s centroid is closest to each object. Hierarchical-based methods work by decomposing a given dataset into a hierarchical structure. They can be divided into bottom-up approaches, which start each data object as a cluster and iteratively merge them to form a hierarchy, and top-down approaches, which start the entire data set as a single cluster and iteratively split it into smaller clusters (Saxena et al., 2017; Han, Pei & Kamber, 2011; Cai et al., 2023; Neto et al., 2019). Grid-based methods, such as STING and WaveCluster (Wang, Yang & Muntz, 1997; Sheikholeslami, Chatterjee & Zhang, 1998), quantize the data space into cells to form a grid structure. Each of the cells is then subjected to the clustering operations. This approach is both efficient and fast because grid-based methods are not affected by the number of data objects. However, the clustering results depend on how the cells of the grid get partitioned.

The density-based clustering method, and specifically the prominent DBSCAN algorithm (Ester et al., 1996; Bhattacharjee & Mitra, 2021), is a type of clustering technique which defines clusters in feature space as dense regions separated by regions of sparser density. The density peak clustering is another type of density-based clustering method that identifies peaks in the data and designates them as cluster centers. Density peak clustering groups data points under the assumption that a data point and its closest high-density neighbors are in the same cluster (Rodriguez & Laio, 2014; Yan et al., 2017; Chen et al., 2020; Hou, Zhang & Qi, 2020). The DBSCAN and similar density-based methods exhibit desirable properties when compared to other clustering methods such as the partitional and hierarchical methods, such as the ability to detect and handle outliers (noise objects), discover clusters of arbitrary and irregular shapes and independence from pre-existing knowledge on the number of clusters present in the dataset.

For algorithms in the density-based clustering method, the core idea behind identifying clusters is modularized into two operations. First, the algorithm devises a method to estimate the region density for a given data object. Such operation helps the algorithm understand whether the data object resides in a region of low density or high density. This information is used as a basis for constructing a cluster in the second operation of the algorithm by expanding to reachable data objects restricted to within the same dense region and assigning a common cluster label to all objects.

The DBSCAN algorithm uses the concept of region density to determine whether a data space consists of a cluster. It uses information from two user input parameters, minPts and ɛ, to determine how dense the space is (“neighborhood”) around a given data object in the dataset. It defines the neighborhood around a data object as dense if at least minPts number of data objects are present within the ɛ units of distance of that data object. The algorithm employs a region querying mechanism on a given data object and determines if the region consists of a cluster based on the density around the object’s neighborhood (Ester et al., 1996; Bushra & Yi, 2021).

However, the DBSCAN exhibits some limitations. The algorithm is dependent on the user-specified ɛ parameter (Bushra & Yi, 2021). Based on the value set for ɛ from the user, the clusters found by the algorithm could vary widely. A higher value of ɛ tends to inadvertently group a set of data objects into a single cluster that should otherwise have been clustered separately. A lower value might misclassify a set of data objects that belong in a cluster as noise if the density in the cluster’s region does not satisfy the preset value of ɛ. When DBSCAN is applied on such a dataset with an inappropriate ɛ as the global parameter, either multiple clusters join into one (a larger value of ɛ), or groups of data objects with a local density less than ɛ are assigned as noise (a smaller value of ɛ) (Bushra & Yi, 2021). This shows the high sensitivity of the DBSCAN algorithm on the user parameter ɛ. For DBSCAN to return an accurate result of cluster labels, the optimal ɛ value needs to be detected.

Another limitation the DBSCAN algorithm faces is during the clustering of intersecting dense regions in the dataspace (Bushra & Yi, 2021; Tran, Drab & Daszykowski, 2013). As the algorithm starts by choosing an arbitrary unlabeled object, its subsequent operation of finding neighborhood core objects and performing expansions could result in different outputs of cluster members where border objects of separate groups of objects are within neighborhoods of each other.

In this article, we introduce a method to programmatically determine the ɛ parameter for a given dataset, based on the distribution data gathered from its k-nearest neighbor (kNN)  (Bhatia, 2010; Fix & Hodges, 1989) computation. The kNN computation returns information on the region density of the data objects in the dataset for a wide range of ɛ distances. We build on the information gathered from the kNN and its distribution to find the optimal ɛ value that returns the most relevant cut-off in terms of identifying the existing clusters in a dataset. Our proposed approach also deals with the DBSCAN algorithm’s unpredictable results where two separate clusters appear relatively close. In such cases where the density region of border objects of one cluster overlaps the density region of another cluster, the original DBSCAN returns different cluster labels for the border objects on different runs of the algorithm. To alleviate this, we provide an efficient re-implementation of the DBSCAN procedure that continuously updates the labels of the border objects to find their appropriate cluster groups. Specifically, the contributions of this article are as follows:

1. Determine the ɛ parameter for a given dataset programmatically so that hyperparameter tuning can be omitted. Doing so removes the burden on the users of the density-based clustering technique and enables the automatic detection of the clusters of a given dataset.

2. Improve upon the accuracy of the DBSCAN algorithm when clustering data found in overlapping density regions by performing additional operations on border objects without increasing the complexity.

3. Verify the reliability and stability of the ɛ parameter computed by the proposed algorithm and demonstrate the superior performance and relatively fast execution time of AutoSCAN through comparative experiments on several datasets.

The rest of this article is organized accordingly. ‘Related Works’ investigates the DBSCAN algorithm and its core idea, as well as other previously proposed density-based approaches. ‘Background’ focuses on our proposed algorithm, AutoSCAN, and its methodology. ‘Proposed Method’ provides an experiment section where validation tests for each of our proposed methods have been conducted and a comparative analysis section to compare our results against other works. Finally, ‘Experiments’ concludes the article by providing an overview of the study.

Related Works

The varied density-based spatial clustering of applications with noise (VDBSCAN) (Liu, Zhou & Wu, 2007) is a method that adopts the plotting mechanism of the objects in the dataset in order to extract clusters with differing densities. Before operating the DBSCAN algorithm, VDBSCAN proposes to compute the distance of the objects in the dataset to their k-nearest neighbor. The objects are sorted by their kNN (k-dist) values and plotted on a graph. The k-dist plot produces sharp changes at suitable values of ɛ. The DBSCAN algorithm is computed for each corresponding values of ɛ. At each run of the algorithm, objects detected in a cluster at ɛi, are marked as clustered at density level i. Marked objects do not participate in further computation of DBSCAN. The two-step process of identifying varied-density clusters based on k-dist is independent of ɛ as an input.

The VDBSCAN algorithm suggests extracting multiple epsilon values for a single dataset to cluster the data objects at different levels of density. As such, it proposes examining the k-nearest neighbor distances of the objects in the dataset to find the required epsilon values. The algorithm, once the kNN values are calculated, sorts the values ascendingly which results in a k-dist plot. The authors then propose identifying one or more regions of the graph with sharp changes as the location for the suitable value of epsilon parameters. Intuitively, a sharp change appearing on the graph suggests a significant population of data objects in the dataset with a common ɛ value, hence a “dense region”, which defines a cluster in terms of density-based clustering methods (Liu, Zhou & Wu, 2007; Bushra & Yi, 2021). However, this approach faces certain challenges. First, the method does not propose a mechanism to programmatically detect such regions that hold a suitable epsilon value. This means, user interference is still needed to correctly cluster a given dataset. Furthermore, as more datasets are examined under this method, it is apparent that (a) the method fails to return a distinct “sharp change” region for datasets with higher number of instances; and (b) datasets that do not exhibit clusters with varying density distributions do not always exhibit significant changes in the k-dist plot.

The study by Tran, Drab & Daszykowski (2013) tackles the limitation of DBSCAN that returns unpredictable cluster labels for data objects in overlapping density regions. The algorithm focuses on the idea of density-reachable chains of objects as a solution. From the definition of density-reachability, border objects could only be present in the density-reachable sets of objects as the last element of the chain. Hence, border objects do not participate in the expansion of clusters. Therefore, the authors propose a new concept, the core-density-reachable objects, where the chains of objects are x1, ..., xn for xi ∈ D and |Nɛ(xi)| ≥ minPts, i ≤ n (all the objects in the chain are core objects). The ExpandCluster is then further revised in order to identify border objects for the corresponding expanded clusters. Due to the implementation of core-density-reachability, the border objects would remain unclassified during the ExpandCluster operation. Therefore, for each border object, the closest core object is identified and clustered to the core-density-reachable chain in which the core object belongs to Tran, Drab & Daszykowski (2013) and Bushra & Yi (2021).

An additional module is introduced to cluster the border objects which were unassigned during the ExpandCluster process. This re-implementation however causes an increase in complexity. For each of the unlabeled border objects, assigning its optimal cluster label requires information about its neighborhood objects. As such, RegionQuery process is run. This leads to an expensive addition to the original algorithm, as the re-introduced RegionQuery procedure could lead to an additional O(n2) cost of execution.

In addition to the abovementioned two algorithms, many ideas have been suggested to improve the parameter detection problem or the misclassification of some objects in the overlapping regions of a cluster in the original DBSCAN algorithm (Bushra & Yi, 2021). Ordering points to identify clustering structures (OPTICS) (Ankerst et al., 1999) was proposed to generate an ordering of a dataset that represents a density-based clustering structure corresponding to a wide range of parameter values. The advantage of this clustering method is that the reachability plot is relatively insensitive to the input parameter ɛ. However, OPTICS does not explicitly assign a cluster to each data object; instead, it holds the information needed for extended DBSCAN to assign cluster relationships to all data objects. Further, OPTICS is good at finding clusters in dense regions, but weak at finding information about clusters in sparse datasets.

The hierarchical density-based spatial clustering (HDBSCAN) algorithm (McInnes, Healy & Astels, 2017; Neto et al., 2019) solved the limitations of OPTICS by using hierarchical clustering methods. This algorithm is based on the concept of OPTICS and applies a mutual reachability graph. The clusters in the graph are the connected components of the ɛ-core objects. HDBSCAN performs DBSCAN for a range of ɛ values and integrates the results to find the clustering that provides the best stability over ɛ.

The density-linked subspace clustering (SUBCLU) algorithm (Kailing, Kriegel & Kröger, 2004) performs density-based clustering by using subspace clustering techniques. When clustering high-dimensional datasets, the DBSCAN algorithm considers all dimensions of each pair of objects to measure the distance and form clusters. By contrast, the SUBCLU algorithm can detect arbitrarily shaped and positioned clusters in subspaces, and it uses DBSCAN as its underlying clustering method. Information about these algorithms is summarized in Table 1.

Table 1 Descriptions of the related algorithms.

Algorithm	Time complexity	Description	
DBSCAN (Ester et al., 1996)	O(n2)	Clusters data by distinguishing dense from sparse regions.	
OPTICS (Ankerst et al., 1999)	Orders dataset that represents a density-based clustering structure for various parameter.	
VDBSCAN (Liu, Zhou & Wu, 2007)	Extracts varying values of ɛ based on k-dist plots and identifies clusters of different densities.	
Revised DBSCAN algorithm to cluster data with dense adjacent clusters (Tran, Drab & Daszykowski, 2013)	Utilizes density-reachable chains to address labeling inconsistencies in overlapping areas.	
AutoSCAN	Determines the ɛ automatically, and efficiently clusters border objects of overlapping density regions.	
HDBSCAN (McInnes, Healy & Astels, 2017)	O(dn2)	Employs hierarchical clustering and mutual reachability graphs to form density clusters.	
SUBCLU (Kailing, Kriegel & Kröger, 2004)	O(2dn2)	Applies monotonicity of density-connectivity sets and subspace information for high-dimensional data clustering.	

Background

In this section, we revise the concept of density-based clustering method (Kriegel et al., 2011; Aggarwal & Reddy, 2014; Sun et al., 2022; Wang & Yang, 2021) and its pioneer approach: the DBSCAN algorithm. Additionally, we mention two previously proposed methods developed on top of the DBSCAN aimed to improve on its accuracy. We study the definition of a “cluster” in terms of density-based method and give an overall description on the building blocks of DBSCAN. Each limitation and disadvantages of these previous methods have also been discussed.

The DBSCAN algorithm

The formal definitions of the clustering model and the DBSCAN algorithm were first introduced together in 1996, on the Knowledge Discovery in Databases (KDD) data mining conference publication (Ester et al., 1996; Schubert et al., 2017). The core idea behind the density-based clustering method assumes that a cluster is a region in a dataspace with a high density of data objects. This clustering model brought unique features different from the earlier mentioned implementations. It introduced the ability to form clusters of irregular shapes, detect outliers in the dataspace, and identify clusters without prior knowledge of the classes present in the dataset. It is a practical algorithm, and the DBSCAN, has been applied in several fields of study such as civil engineering, chemistry, spectroscopy, social sciences, medical diagnostics, remote sensing, computer vision, automatic identification systems (AIS) and anomaly detection. Its successful implementation on real-world applications has led it to receive the Special Interest Group on KDD test-of-time award (Schubert et al., 2017).

The clustering notion of the DBSCAN algorithm is based on identifying densely populated regions of data space separated by sparser regions and interpreting them as clusters. In addition, DBSCAN uses two input parameters ɛ and minPts, to identify the density estimation of a region surrounding a particular data object. The algorithm uses these two parameters to estimate the density of a particular data object’s local region. Moreover, ɛ refers to the radius of an object’s local region (neighborhood), whereas minPts is the minimum number of data objects required within that radius in order to be a cluster. Therefore, for a given dataset D, the DBSCAN algorithm calculates the local density of an object xi, xi ∈ D as the total number of objects in its ɛ-neighborhood (i.e., cardinality of Nɛ(xi)) (Ester et al., 1996; Bushra & Yi, 2021), where (1) Nɛxi=xj∈D:∀j,distxi,xj<ɛ

Each data object in a given dataset are classified as either a core object, border object or noise depending on the denseness of the surrounding region (neighborhood) the object appears in. Core objects are those where their ɛ-neighborhood contains at least minPts number of objects (i.e., |Nɛ(xi)| ≥ minPts, for some xi ∈ D). These objects indicate that, for the given values of ɛ and minPts, their neighborhood objects (along with the core objects themselves) form a cluster. The number of objects in border objects’ ɛ-neighborhood is less than the given minPts; however, for a border object xj, a core object xi exists where xj ∈ Nɛ(xi). The third type is noise objects. These objects do not have at least minPts number of objects in their minPts-neighborhood and are not members of any core object’s neighborhood (Ester et al., 1996; Bushra & Yi, 2021). The DBSCAN further makes definitions regarding reachability of data objects in the dataset. The definitions are presented as follows.

A. Directly density reachability An object xi ∈ D is directly density-reachable from an object xj ∈ D for a given ɛ and minPts if: (a) xi is a member of the ɛ-neighborhood of xj, and(b) xj is a core object (Ester et al., 1996; Bushra & Yi, 2021): (2) DirReachxi,xj⇔xi∈Nɛxj∧Nɛxj≥ minPts.

B. Density reachability An object xi ∈ D is density-reachable from an object xj ∈ D with respect to a given ɛ and minPts if a series of objects o1, …, on, exists, where o1 = xj and on = xi such that oi+1 is directly-density reachable from oi (Ester et al., 1996; Bushra & Yi, 2021): (3) Reachxi,xj⇔∃oi,…,on∈D:o1=xj∧on=xi∧∀i∈1…n−1:DirReachoi,oi+1

By this definition (Ester et al., 1996; Bushra & Yi, 2021), two border objects that appear in the same cluster might not be density reachable from each other due to the core object constraints set for the series of objects o1, ..., on. However, they are directly density-reachable to a common core object. Density-connectivity defines this property.

C. Density connectivity An object xi ∈ D is density-connected to an object xj ∈ D with respect to a given ɛ and minPts if there is another object o such that both xi and xj are density-reachable from o with respect to ɛ and minPts (Ester et al., 1996; Bushra & Yi, 2021): (4) Connectxi,xj⇔∃o∈D: Reacho,xi∧ Reacho,xj

Based on these notions, DBSCAN defines a cluster as a maximal set of density-connected objects concerning density-reachability. That is, a cluster C with respect to ɛ and minPts is a nonempty subset of D such that: a. ∀xi, xj: if xi ∈ C and xj is density-reachable from xi with respect to ɛ and minPts, then xj ∈ C, and b. ∀xi, xj ∈ C: xi is density-connected to xj with respect to ɛ and minPts. Finally, noise objects are a set of objects in the dataset that do not belong to any of its clusters.

The original DBSCAN algorithm was successful in identifying clusters with arbitrary shapes. This feature is appreciated, especially in spatial databases where clusters can be spherical or even straight, bending, and other shapes. The characteristics of DBSCAN to identify regions with high density that are separated by sparser spaces allows its clusters’ shapes to be determined by the dataset under examination as opposed to other clustering methods, such as the partitional clustering k-means, which always assumes a spherical shape for its clusters. In addition, DBSCAN can identify isolated data objects and is able to assign them as noise.

Proposed Method

The original density-based clustering method, DBSCAN, is a pioneering density-based clustering algorithm capable of extracting clusters of irregular shape and identifying outlier data objects that do not belong to any cluster. While this implementation provides good results, especially in spatial datasets, it does however exhibit two important limitations. First is DBSCAN’s reliance and sensitivity to its input parameters. The definition of a cluster in DBSCAN requires the computation of density regions of the data objects present in the dataset. This operation, performed by the RegionQuery module in the procedure, is dependent upon the values set for the parameters ɛ and minPts. As such, an accurate tuning of values is required to correctly identify the clusters present in the given dataset. However, as the DBSCAN algorithm takes these parameters from its users, the final clustering output can be varying. If a lower ɛ value is set, a mis-classification of data objects as outliers (noise objects) is exhibited. While a higher ɛ value has the effect of merging data objects from separate clusters into a single group. To alleviate this, we propose a method to automatically detect the ɛ parameter based on the density distribution of a given dataset.

The second limitation of the DBSCAN algorithm is demonstrated when clustering data objects found in overlapping density regions of separate clusters. During the recursive process of the ExpandCluster module, the DBSCAN finds all the density-connected data objects from core objects found in the dataset. This operation terminates each iteration when a border object is detected, and further density-connected data objects could not be found. All the data objects collected during this phase (“seeds”) are subsequently assigned similar cluster label. However, for border objects located in a dataspace where there are overlapping density regions of separate clusters, the ExpandCluster does not guarantee a correct label assignment. The ExpandCluster assigns cluster labels for the border objects appearing in such regions the cluster label of the first core object it operates on. This leads to unpredictable and inaccurate results as the first core object to find each border object is not always its correct cluster member. Additionally, the unpredictability is presented when multiple iterations of the DBSCAN algorithm is performed with the data objects visited in random order. This causes the cluster assignments of the objects in overlapping density regions to alter on each run of the algorithm. Our method to accurately cluster data objects in overlapping density regions solves this problem by introducing a new ClosestCore concept within the ExpandCluster module that keeps track of the core objects visiting the objects in overlapping density regions and iteratively updating their cluster label.

Figure 1 Pipeline of our proposed algorithm AutoSCAN.

Our first method is implemented as a preprocessor before the recursive operation begins. It takes the dataset from the user and returns the computed epsilon to the revised clustering algorithm. Our second method is implemented within the ExpandCluster operation to compute border objects’ cluster label.

As discussed, the two proposed algorithms implement additional modules into the original DBSCAN’s process. The overview of the proposed algorithms within the DBSCAN operation is illustrated in Fig. 1. The first method proposed to automatically detect epsilon value operates before the recursive process of the original DBSCAN algorithm begins. This proposed method first takes the dataset and assigns the minPts required for the clustering operation based on its feature size. Next it calculates the k-nearest neighbors’ distances, calculate the frequency distribution, applies a curve-fitting function and determines the optimal region for the dataset’s ɛ value from the output curve. After this operation is completed, the computed parameters, the ɛ and minPts are passed onto the clustering algorithm’s recursion to start its selection of random data object and perform the RegionQuery. The second efficient method proposed to cluster data in overlapping regions is implemented within the recursive procedure. Specifically, in the ExpandCluster operation, this method works by similarly collecting all the density-connected data objects from a given core object. Next, it proceeds to compute the distance between the border object within the seeds and the core object to determine their proximity. This additional attribute introduced for the border objects is used to recursively update the cluster label of each border object as a closer core object within their neighborhood is detected. When the recursive process is completed, all the border objects will belong to their closest core objects’ cluster labels.

Automatic detection of epsilon parameters

The first proposed method utilizes information from a dataset’s kNN distances and the B-spline curve fitting function. Figure 2 shows the overview for the first proposed algorithm.

Figure 2 Pipeline of our first proposed method.

Our first method initiates by determining the k-nearest neighbor distances for each data object in the dataset. Next it generates a B-spline curve from the frequency distribution of the k NN values to identify the dataspace’s density distribution. The optimal epsilon is detected in the final iteration step through the curve fitting values.

k-nearest neighbors

The k-nearest neighbor algorithm (Bhatia, 2010; Fix & Hodges, 1989), as a supervised learning algorithm, can be used to solve both the classification and regression problem. This assumes that similar things exist in proximity (i.e., similar things are near to each other). This assumption resonates with the concept of density-based clustering, which assigns data objects densely stacked together within a given radius of a dataspace data space as a single cluster.

To perform the kNN, first initialize k to a specific number of neighbors. The value of k set in the algorithm influences the frequency distribution, which affects the number of resulting clusters. Therefore, we determine it based on the feature size of the dataset (i.e., k = 2 × d, d = featuresize) by referring to the other studies’ methods (Ester et al., 1996). Next, for each data object in the dataset, compute their kth closest object based on a distance similarity function. The Euclidean Measure is commonly used as the function to compute distance similarity. Given that the minPts value is set to k, the distance value to the kth-nearest neighbor computed by kNN for each individual object corresponds to the minimum ɛ value required for that data object to be defined as a core object. Thus, for the range [min, max] of a set whose elements are the distance values to the kth-nearest neighbor for each object, if min ≤ ɛ ≤ max is set as the cut-off ɛ value, any data object x ∈ D with kNN(x) ≤ ɛ is considered a core object. Further, under the definition of the DBSCAN clustering algorithm, any data object y ∈ D with kNN(y) > ɛ is considered a border object or noise, where kNN(⋅) represents the distance value to the kth-nearest neighbor of the given object.

Once the kNN distances of the data objects is calculated, the proposed method uses the data to compute the frequency distribution of the distances. This determines the density distribution of the dataset. By observing the kNN frequency distribution, it can be deduced that peak areas of the distribution correspond to an ɛ value with large population of data objects. Hence, such data objects should be regarded as core objects (i.e., the optimal ɛ value is greater or equal to the ɛ value in such region). Whereas low population within the distribution indicates minimal number of data objects with such density. Therefore, for a given dataset, the optimal ɛ cut-off value corresponds to the region in the frequency distribution in which the population of data objects decelerates from a peak area to a “valley” (low population of data objects). Additionally, as the frequency distribution of a given dataset could possibly contain multi-modality in terms of peaks (ɛ value with large population), our findings suggest that setting the cut-off region to the last deceleration point returns the best clustering outcome. This is similar to density peak clustering (Rodriguez & Laio, 2014; Yan et al., 2017; Chen et al., 2020; Hou, Zhang & Qi, 2020) in that it calculates the kNN distance, but it differs in that it analyzes the entire frequency distribution to find the optimal ɛ value, rather than calculating the density of each data point.

B-spline curve fitting

Curve fitting is the process of constructing a curve, or mathematical function, that has the best fit to a series of data objects, possibly subject to constraints (Arlinghaus, 1994; Kolb, 1984). Curve-fitting can either involve interpolation, where an exact fit to a data is required, or smoothing, in which a “smooth” function is constructed that approximately fits the data. For our purposes, we employed our series of frequency distribution data into a smoothing function in order to iterate through the data and identify the appropriate ɛ cut-off region.

The cubic B-spline curve function (Gordon & Riesenfeld, 1974) is used for curve-fitting the series of data objects obtained from the frequency distribution of the kNN of the dataset. A B-spline curve is defined as a linear combination of control points pi and B-spline basis functions Ni,k(t) given by: (5) rt= ∑i=0npiNi,kt,n≥k−1,t∈tk−1,tn+1

In this context, the control points are called de-Boor points. The basis function Ni,k(t) is defined on a knot vector: (6) T=t0,t1,…,tk,tk+1,…,tn−1,tn,tn+1,…,tn+k,

where there are n + k + 1 elements, i.e., the number of control points n + 1 plus the order of the curve k (3 for cubic B-spline). Each knot span ti ≤ t ≤ ti+1 is mapped onto a polynomial curve between two successive joints r(ti) and r(ti+1).

Iterating through the density distribution

Given a window size w, the optimal ɛ region of the dataset is identified by moving through the kNN graph computed in previous step to detect the curve or “valley” that corresponds to best cluster result. In order to detect such region, initialize the window from the end of the graph (i.e., start the window from the maximum ɛ value) and iterate backwards. At each instance, calculate the mean function of ɛ values within w and keep iterating until the mean starts increasing. “Window size” refers to a segment of plot points on the kNN graph. By intuition, as the segment of w increases, the sensitivity in terms of “mean” value decreases. The kNN B-spline curve is set to generate min(len(p)∗2, n) points, where n the number of instances in the dataset; therefore, w moves through these points to calculate the mean values of each window and determine a drop point. The optimal ɛ is determined to be the computed mean of the curve region with the steepest incline. Hence, once the optimal ɛ region is determined, the proposed algorithm sets the value of ɛ to the mean of the values in the ɛ region.

Finally, the procedure moves on to the recursive DBSCAN procedure with the computed ɛ and minPts calculated from the feature size as input.

Efficient clustering of data in overlapping density regions

The second proposed algorithm presented in this article devises an implementation to accurately cluster data objects that lie in regions with overlapping neighborhoods of core objects with different cluster labels. The procedure is implemented within the ExpandCluster operation in the DBSCAN algorithm. The overall flow of this operation is illustrated on Fig. 3.

Figure 3 Pipeline of our second proposed method.

Our second method is operated within the ExpandCluster module. The revised ExpandCluster computes the new ClosestCore value for the border objects it detects. During the module’s iteration through core objects, the ClosestCore of each border object is updated accordingly.

ClosestCore computation

To accurately cluster each data object, the proposed algorithm introduces an additional concept, the ClosestCore Object. During the implementation of the original ExpandCluster, all the density-connected objects from a single core object x ∈ D are gathered and denoted as “seeds”. While the original DBSCAN algorithm assigns each object to the cluster label of the core object x without further investigation, the proposed algorithm performs advance computation to accurately cluster all the data objects. The proposed algorithm filters the data objects y that are denoted as borders. For each border, DirectReach(x, y) computes the distance between itself (y) and the core object x. This distance is denoted as the ClosestCore Object for object y if the distance is lower or equal to the current ClosestCore Object distance.

____________________________ Algorithm 1 ExpandCluster__________________________________________________________________________________________   1:  D ← Dataset   2:  x ← Data Object to Expand i  3:  clusterID ← Current Cluster ID   4:  procedure EXPANDCLUSTER(D,x,clusterID)   5:       seeds = RegionQuery(D,x)   6:       if seeds.size < minSamples then  7:            x.type = NOISE  8:            return false  9:       else 10:            x.type = CORE 11:            for j in seeds do 12:                  x.label = clusterID 13:            end for 14:            while seeds.size > 0 do 15:                  xi = seeds.pop() 16:                  neighbors = RegionQuery(D,xi) 17:                  if neighbors.size ≥ minSamples then 18:                       xi.type = CORE 19:                       for n in neighbors do 20:                            if n.type == UNCLASSIFIED then 21:                                  n.label = clusterID 22:                                  seeds.append(n) 23:                            end if 24:                            if n.type == BORDER then 25:                                  n.ClosestCore = min(n.ClosestCore, dist(n, xi)) 26:                                  n.label = n.ClosestCore.label 27:                            end if 28:                       end for 29:                  else 30:                       xi.type = BORDER 31:                       xi.ClosestCore = dist(xi, x) 32:                  end if 33:            end while 34:       end if 35:  end procedure_________________________________________________________________________________________

(7) ClosestCorey=distx,y,ifClosestCoreyisUNDEFINEDminClosestCorey,distx,y.

As the clustering algorithm is a recursive process, the ClosestCore distance continuously updates depending on the proximity of each core object to the border object y. The cluster label of the border object is finally determined from the cluster label for the core object determined from the ClosestCore object. This ensures that the final label for all the border objects in the dataset hold the cluster label of their ClosestCore object.

ExpandCluster re-implementation

The original structure of the ExpandCluster operation first starts by collecting a core object x ∈ D as a starting object and performs iterations of RegionQuery runs on x’s neighborhood data objects to identify further core objects. When core objects are identified, the module repeats the process of RegionQuery calls on the neighborhood core objects. All the data objects collected from such recursive procedure are then saved as “seeds” and assigned similar cluster labels along with the core object x. This procedure helps the original DBSCAN collect all the density-connected data objects from x. However, for a border object y:y ∈ D∧Connect(x, y) found during this operation, the original ExpandCluster implementation does not perform further query beside assigning it the same cluster label. If border object y exists in an overlapping density region of individual clusters, it will likely be misclassified because it may be closer to a core object in another cluster. Therefore, additional queries are required for efficient clustering. In this light, we propose an algorithm to update the information of the core object closest to the border object. Our proposed algorithm computes a distance measurement between the border object y and the density-connected core object x. This distance measurement is then stored as the ClosestCore value of the border object y (i.e., y.ClosestCore = dist(x, y)). Furthermore, during an ExpandCluster operation of another core object z, if border object y is detected as a density-connected object to z, y’s ClosestCore value is updated based on Eq. (7). Finally, once all the ExpandCluster calls have been made, each border object would belong to each of their corresponding core objects’ cluster labels. The pseudocode for the revised ExpandCluster module is presented in Algorithm 1 .

Experiments

In this section, we discuss about the datasets used for the experiments, the evaluation method for validation tests, the algorithms conducted and analyze the output of each algorithm. Validation tests and comparative analysis are conducted for our proposed algorithm. For the validation tests, we perform separate evaluations on our two proposed methods to understand and analyze the output of each operation. In the validation section, we run our proposed algorithms on synthetic datasets with pre-defined labels and show that they return with expected outputs. Next, we perform comparative analysis of our proposed algorithm against other relevant works in the literature; along with the algorithms mentioned in Section ‘Related Works’.

For our experiments, we gathered four collections of datasets. We use the datasets in Fundamental Clustering and Projection Suite (FCPS) (Thrun & Ultsch, 2020) to make validation tests for our proposed algorithm. We also run our proposed algorithm along with other target algorithms on real-world datasets to test the computational time required for each method. We compare the accuracy and time complexity of our proposed algorithm against the original DBSCAN algorithm as well as the algorithm discussed on Section ‘Related Works’. All experiments reported here were conducted on two Intel Xeon CPU E5-2695 processors clocked at 2.10 GHz.

Datasets

Fundamental clustering and projection suite

The Fundamental Clustering and Projection Suite (FCPS) (Thrun & Ultsch, 2020) contains 10 synthetic datasets; namely “Atom,” “GolfBall,” “Hepta,” “Chainlink,” “EngyTime,” “LSun,” “Target,” “Tetra,” “TwoDiamonds,” and “WingNut”. Table 2 describes the instances and dimensions of each of the datasets used for the experiment. Each one was built to challenge clustering algorithms based on different criteria. Each dataset has specific criteria such as the lack of linear separability, class spacing differences, outlier presence, and others. Hence, the table also details these criteria and problems clustering algorithms could face for each dataset in the FCPS.

Table 2 Details of the datasets used in the fundamental clustering and projection suite datasets (FCPS).

Name	Instances	Feature size	Description	
Atom	800	3	Different inner class distances	
Chainlink	1,000	3	Linearly not separable	
GolfBall	4,002	3	No cluster structure	
Hepta	212	3	Different inner class distances	
LSun	400	2	None	
Target	770	2	Presence of outliers	
Tetra	400	3	Small inter class distances	
TwoDiamonds	800	2	Touching classes	
WingNut	1,016	2	Density variation within classes	

The components of the FCPS each have cluster labels for their data objects. This information was used to compare the data objects’ cluster labels to the labels generated by the clustering algorithms during the experiment. The F-score and Adjusted Rand Index (ARI) measure used the cluster labels to measure the agreement between the labels formed by the clustering algorithms and the labels from the FCPS.

scikit-learn synthetic datasets collection

The scikit-learn synthetic datasets (Pedregosa et al., 2011) are a set of six 2-dimensional datasets, each with 1,500 data instances generated as benchmark datasets to test the capability of various clustering algorithms. Like the FCPS, each dataset in this collection exhibits special properties that emulates a real-world dataset, aimed to test a clustering algorithm’s capacity in several manner. Each dataset is provided with a class label, which is used in our experiment to test and compare the accuracy of ours and the various other clustering algorithms. The features and properties of each dataset is presented in Table 3.

Table 3 Details of the datasets used from the scikit-learn clustering benchmark collection.

Name	Instances	Feature size	Description	
noisy_circles	1,500	2	Linearly not separable	
noisy_moons	
aniso	Anisotropicly distributed data	
blobs	Density uniformly distributed	
varied	Blobs with varied variances	
no_structure	Presents no cluster structure	

Geo-tagged twitter dataset

Junjun Yin from the National Center for Supercomputing Application (NCSA) collected the real-world dataset for the experiments (Götz & Bodenstein, 2015). The dataset was obtained through the free Twitter streaming API. The original collection contains exactly 1% of all geo-tagged tweets from the United Kingdom in June 2014 and has about 16.6 million tweets. The subset of the dataset used for the experiments was generated by filtering the dataset to on the first week of June. This filtered dataset contains 3,704,351 instances. Furthermore, in order to analyze how the proposed clustering algorithms perform as the number of instances in the dataset increases, experiments on the dataset were run on several partitions of the dataset, (i.e., at 1,000, 10,000, 50,000, 100,000, 1,000,000 and 2,000,000 instances). The full and filtered dataset can be found on B2SHARE.

Real-world multi-dimensional datasets

Furthermore, we have collected eight real-world datasets of varying feature sizes to better understand the effect of dimensionality in terms of time performance. All eight of the datasets have been collected from the UCI Machine Learning Repository and can be found here Dua & Graff (2019). Table 4 presents the description, feature size and number of instances of the datasets collected.

Table 4 Description of the UCI real-world multi-dimensional datasets.

Name	Instances	Feature size	Description	
Iris	150	4	Types of iris plants	
Travel Review	980	10	Reviews on destination areas	
Wine	178	13	Chemical analysis of winescreated in Italian regions	
MFCC	7,195	22	Audio records of different	
Anuran species calls	
Ecoli	336	8	Protein localization sites	
Yeast	1,484	8	
Glass	214	10	Glass identification datain terms of oxide content	
WDBC	569	30	Breast cancer Wisconsin (diagnostic) dataset	

Validation tests

Automatic detection of ɛ parameters

To perform validation tests on our first proposed method to automatically determine ɛ parameter, the FCPS collection of datasets was used. The pre-defined cluster labels available from this synthetic collection of data was regarded as the optimal cluster labels for the objects in the datasets. Thus, the cluster labels were useful to analyze the accuracy obtained from the proposed method. To show a quantitative analysis of the cluster accuracy, the Adjusted Rand Index (ARI) (Rand, 1971; Hubert & Arabie, 1985) measure was implemented when comparing the output of the algorithm against the cluster labels.

The Rand Index (RI) (Rand, 1971; Pedregosa et al., 2011) computes a similarity measure between two clusterings by considering all pairs of samples and counting pairs that are assigned in the same or different clusters in the predicted and true clusterings: (8) RI=number of agreeing pairsnumber of pairs.

This raw RI is then adjusted for chance into the ARI score we used as our accuracy measure using the following scheme: (9) ARI=RI−ExpectedRImaxRI−ExpectedRI.

This measure of adjustment ensures the score has a value close to 0.0 for random labeling independently of the number of clusters and samples and exactly 1.0 when the clusterings are identical.

To validate the results of our first proposed algorithms, ten ɛ values between [max, min] range of the kNN were randomly generated for each of the datasets in the FCPS. We then used the ɛ parameters to run the DBSCAN algorithm and measured the accuracy of each value. Table 5 shows each of the ɛ value used for the datasets and the ARI accuracy score. The values that returned the best DBSCAN cluster results have been highlighted.

Table 5 Validation tests results of our first proposed method using the datasets from FCPS.

The text in bold indicates the ɛ values that best cluster the respective dataset (i.e., these ɛ values return the highest ARI score). The values in underline indicate the ɛ results automatically detected by our proposed method.

Atom	Chainlink	GolfBall	Hepta	Lsun	Target	Tetra	TwoDiamonds	WingNut	
epsilon	ARI	epsilon	ARI	epsilon	ARI	epsilon	ARI	epsilon	ARI	epsilon	ARI	epsilon	ARI	epsilon	ARI	epsilon	ARI	
0.768	0.995	0.115	0.04	0.077	0	0.491	0.995	0.215	0.5	0.1534	0.578	0.584	0.068	0.158	0.512	0.215	0.965	
0.796	0.995	0.127	0.06	0.085	0	0.512	0.995	0.227	0.839	0.1695	0.582	0.593	0.081	0.164	0.622	0.227	0.988	
0.806	0.997	0.131	0.08	0.088	0	0.519	0.995	0.231	0.898	0.1750	0.598	0.596	0.087	0.166	0.779	0.231	1.0	
0.856	1.0	0.163	0.667	0.109	0	0.5724	1.0	0.251	0.90	0.218	0.807	0.620	0.135	0.182	0.779	0.238	1.0	
0.884	1.0	0.165	0.73	0.11	0	0.5744	1.0	0.265	1.0	0.2195	0.807	0.621	0.161	0.1817	0.779	0.265	1.0	
0.922	1.0	0.181	0.992	0.121	0	0.5864	1.0	0.2808	1.0	0.241	1.0	0.633	0.665	0.1823	0.779	0.281	1.0	
0.952	1.0	0.194	1.0	0.129	1.0	0.623	1.0	0.294	1.0	0.258	1.0	0.648	0.665	0.1903	0.779	0.294	1.0	
1.0006	1.0	0.207	1.0	0.1375	1.0	0.658	1.0	0.315	1.0	0.2809	1.0	0.657	0.665	0.197	0.5	0.315	0	
1.124	1.0	0.268	1.0	0.179	1.0	0.746	1.0	0.356	1.0	0.3567	1.0	0.696	0.665	0.207	0.5	0.368	0	
1.175	1.0	0.289	1.0	0.193	1.0	0.782	1.0	0.369	1.0	0.39	1.0	0.712	0.389	0.234	0.5	0.389	0	

From the results collected, our first proposed method was successfully able to identify the ɛ value within each of the dataset’s optimal regions (i.e., the region where the ɛ parameters return the highest clustering result.) While for seven of the datasets the algorithm was successful in finding the ɛ value with a perfect clustering accuracy (ARI = 1.0), for the “Tetra” and “TwoDiamonds” datasets, lower accuracy is reported. However, this only suggests that the original DBSCAN algorithm had limitations in perfectly clustering these datasets. Upon closer inspection, the “Tetra” and “TwoDiamonds” datasets exhibited data objects in overlapping density regions of separate clusters, and hence had misclassified some data objects into different cluster labels.

Efficient clustering of data in overlapping density regions

To validate the results of our second proposed methods, we compare the results found from our proposed algorithm with the results of the original DBSCAN algorithm. The FCPS dataset is used with ARI as an accuracy measure. From our validation results reported on Table 6, we are able to show our proposed method, at worst identifies the same cluster labels as that of the original DBSCAN algorithm. Our proposed algorithm produced more accurate results with datasets containing data objects in overlapping density regions, specifically the “Tetra” and “TwoDiamonds” datasets. The original DBSCAN algorithm often misclassified these objects into different cluster labels. However, because we performed additional operations on the border objects in these regions, we were able to correct the misclassified data objects and improve the accuracy. Specifically, by using our algorithm, all the data objects in the “TwoDiamonds” dataset could be clustered perfectly. Further, our algorithm provides better clustering results for the “Tetra” dataset. The difference in clustering results between the original DBSCAN algorithm and our proposed algorithm on the “TwoDiamonds” dataset is shown in Fig. 4. In this dataset, individual clusters are visually identifiable, and overlapping density regions exists; therefore, the improvements can be identified visually and efficiently.

Table 6 Validation results of our second proposed method showing accuracy improvements when compared with DBSCAN algorithm.

Datasets	Parameters	Accuracy score (ARI)	
Name	Instances	minPts	ɛ	DBSCAN	Proposed (B)*	
Atom	800	6	0.86	1	1	
Chainlink	1,000	6	0.207	1	1	
GolfBall	4,002	6	0.138	1	1	
Hepta	212	6	0.5864	1	1	
LSun	400	4	0.356	1	1	
Target	770	4	0.28	1	1	
Tetra	400	6	0.648	0.665	0.914	
TwoDiamonds	800	4	0.1903	0.779	1	
WingNut	1,016	4	0.238	1	1	

Figure 4 Comparative results of the TwoDiamonds dataset between the original DBSCAN algorithm (A) and our proposed AutoSCAN algorithm (B).

From the results of the DBSCAN algorithm (A), we can see a misclassification of data objects around the border regions of the left “diamond”. This occurs due to overlapping of density regions of the border objects in the left “diamond” with the density regions of the objects in the right “diamond”. The figure on the right (B), a result generated from our proposed algorithm, solves this problem.

Comparative analysis

The comparative experimental results of our proposed algorithm against previous density-based clustering algorithms is reported here. We perform accuracy tests using the FCPS and scikit-learn dataset collection. For quantitative analysis of the algorithms’ accuracy, we have implemented additional accuracy measure, the F-Score, to compare the output of the algorithms against the cluster labels of the synthetic datasets.

The F-score, also called the F1-score, is the measure of a model’s accuracy on a dataset. The F-score is a way of combining the precision and recall of the model, and it is defined as the harmonic mean of the model’s precision and recall (Pedregosa et al., 2011). The value of F-score ranges from 0 to 1, with a perfect model registering an F-score of 1. F1=21recall×1precision

=2×precision×recallprecision+recall

(10) =tptp+12fp+fn.

The time performance tests in this section are conducted using the real-world geo-tagged and multi-dimensional collections of datasets. We report and analyze the running times of the algorithms in seconds (sec.). The algorithms used during time performance tests were written in C++ and executed through the GNU GCC compiler. In the next section, we discuss about the various target algorithms we for comparison against our proposed method.

Target algorithms

In total we have collected seven clustering algorithms to compare against our proposed method. Along with the original DBSCAN algorithm, six of the methods are density-based, while we also include experimental results from the k-means clustering algorithm to analyze our results against a partitioning-based clustering method. To analyze the results of our automatic ɛ detection method, we chose the OPTICS, HDBSCAN and VDBSCAN algorithms as target methods. The VDBSCAN method which uses the k-dist plot as a measure for users to find an ɛ value has also been discussed on Table 1. OPTICS (Ankerst et al., 1999) and HDBSCAN (McInnes, Healy & Astels, 2017) algorithms were chosen due to their ability to detect density-based clusters without requiring the users help to determine the density structure of a given dataset. For OPTICS, we use the xi-clustering algorithm introduced in their original study. Additionally, the clustering method introduced by Tran, Drab & Daszykowski (2013) is used during our experimental analysis as it proposes a method to cluster data with dense adjacent clusters. During our experimental analysis we show that their improved method comes at a time performance cost while our proposed method provides an efficient approach instead. The SUBCLU (Kailing, Kriegel & Kröger, 2004) algorithm previously proposed as a density-connected subspace clustering for high-dimensional data is used primarily in our time performance analysis experiments. We analyze the results of SUBCLU against our proposed method when using the multi-dimensional collection of real-world datasets.

Table 7 Accuracy tests of AutoSCAN algorithm against clustering algorithms proposed to improve the clustering quality of DBSCAN.

The scores of the algorithm with the best accuracy for each dataset are highlighted in bold.

Algorithms	Datasets	Clusters found	Coverage %	Accuracy	
	Name	Size	Feature			F-1 Score	ARI	
Automatic Detection of Epsilon Parameters and Efficient Clustering of Data in Overlapping Density Regions (AutoSCAN)	Atom	800	3	2	100	1	1	
Chainlink	1,000	3	2	100	1	1	
GolfBall	4,000	3	1	100	1	1	
Hepta	212	3	7	100	1	1	
LSun	400	2	3	100	1	1	
Target	770	2	2	98.44	1	1	
Tetra	400	3	4	94	0.939	0.914	
TwoDiamonds	800	2	2	100	1	1	
WingNut	1,016	2	2	100	1	1	
noisy_circles	1,500	2	2	100	1	1	
noisy_moons	2	2	100	1	1	
varied	2	3	95.87	0.97	0.94	
aniso	2	3	99.07	0.99	0.98	
blobs	2	3	100	1	1	
no_structure	2	1	100	1	1	
Ordering Points to Identify the Clustering Structure (OPTICS)	Atom	800	3	2	98.875	0.988	0.978	
Chainlink	1,000	3	2	100	1	1	
GolfBall	4,000	3	1	100	1	1	
Hepta	212	3	7	100	1	1	
LSun	400	2	3	100	1	1	
Target	770	2	2	98.44	1	1	
Tetra	400	3	4	94.75	0.925	0.8766	
TwoDiamonds	800	2	2	100	0.941	0.779	
WingNut	1016	2	2	100	1	1	
noisy_circles	1,500	2	2	100	1	1	
noisy_moons	2	2	100	1	1	
varied	2	3	80.6	0.8	0.71	
aniso	2	3	96.4	0.96	0.94	
blobs	2	3	100	1	1	
no_structure	2	1	100	1	1	
Hierarchical Density Based Clustering (HDBSCAN)	Atom	800	3	2	100	1	1	
Chainlink	1000	3	2	100	1	1	
GolfBall	4000	3	3	100	–	–	
Hepta	212	3	7	100	1	1	
LSun	400	2	3	100	1	1	
Target	770	2	3	100	1	1	
Tetra	400	3	4	91.7	0.925	0.8766	
TwoDiamonds	800	2	3	98	0.99	0.9603	
WingNut	1,016	2	4	99.8	0.99	0.996	
noisy_circles	1,500	2	2	100	1	1	
noisy_moons	2	2	100	1	1	
varied	2	3	89.8	0.88	0.85	
aniso	2	3	96.67	0.97	0.98	
blobs	2	3	100	1	1	
no_structure	2	1	100	–	–	
Varied Density Based Spatial Clustering of Applications with Noise (VDBSCAN)	Atom	800	3	2	100	1	1	
Chainlink	1,000	3	2	100	1	1	
GolfBall	4000	3	1	100	1	1	
Hepta	212	3	7	60.38	0.604	0.3746	
LSun	400	2	3	99.8	0.9975	0.9973	
Target	770	2	2	98.44	1	1	
Tetra	400	3	4	93	0.7629	0.8022	
TwoDiamonds	800	2	1	100	0.5	–	
WingNut	1,016	2	2	97.83	0.978	0.957	
noisy_circles	1,500	2	2	99.73	0.99	0.995	
noisy_moons	2	2	99.6	0.99	0.992	
varied	2	3	95.6	0.724	0.55	
aniso	2	2	99.34	0.66	0.567	
blobs	2	3	100	1	1	
no_structure	2	1	100	1	1	

Accuracy results

For the accuracy tests we show experimental results of AutoSCAN (our proposed algorithm), the original DBSCAN, the method proposed by Tran, Drab & Daszykowski (2013) OPTICS, HDBSCAN, SUBCLU, VDBSCAN and k-Means. We ran the experiments using the FCPS and scikit-learn synthetic datasets. Meanwhile, for time performance tests, the original DBSCAN, OPTICS, SUBCLU and the method by Tran, Drab & Daszykowski (2013) were used to compare against AutoSCAN. The results are reported in the next sections. Tables 7 and 8 show the accuracy results of the algorithms. We used both F-score and ARI as measurements; additionally, the number of clusters and coverage percentage are reported. To clarify the performance comparison, the scores of the algorithm with the best accuracy for each dataset are highlighted in bold. From the experiments, our proposed method, AutoSCAN, reported the superior clustering results in terms of coverage and accuracy measures (both ARI and F-Score). The HDBSCAN, VDBSCAN, and SUBCLU algorithms had lower accuracy for relatively large numbers of datasets compared to other density-based clustering algorithms. While the original DBSCAN algorithm and OPTICS and Tran, Drab & Daszykowski (2013) also reported a considerably high accuracy tests, the AutoSCAN algorithm was able, in its worst cases, match their results. Specifically, for the “Tetra” and “TwoDiamonds” datasets and the “varied” and “aniso” datasets, our proposed algorithm provided increased clustering quality. These datasets are similar in that they contain many data objects in regions of overlapping density between individual clusters; we find that our second proposed algorithm can efficiently cluster these objects. Tran, Drab & Daszykowski (2013) also reported the same accuracy as that of AutoSCAN for the four datasets; however, they also found that some data objects were misclassified as noise in datasets such as “Atom” and “Hepta”. K-means was the only algorithm that perfectly clustered the “Tetra” dataset. This is because a large number of data objects are present in the overlapping density regions between individual clusters in this dataset, as a result of which the density-based clustering algorithm is relatively more likely to misclassify them. However, the k-means algorithm had a 100% coverage rate across all dataset runs, and it failed to find irregularly shaped clusters and identify noise objects.

Table 8 Accuracy tests of AutoSCAN algorithm against clustering algorithms proposed to improve the clustering quality of DBSCAN.

The scores of the algorithm with the best accuracy for each dataset are highlighted in bold.

Algorithms	Datasets	Clusters found	Coverage %	Accuracy	
	Name	Size	Feature			F-1 score	ARI	
Density-based Spatial Clustering on Applications with Noise (DBSCAN)	Atom	800	3	2	98.875	0.988	0.978	
Chainlink	1,000	3	2	100	1	1	
GolfBall	4,000	3	1	100	1	1	
Hepta	212	3	7	99.53	0.995	0.9942	
LSun	400	2	3	100	1	1	
Target	770	2	2	98.44	1	1	
Tetra	400	3	4	94	0.832	0.665	
TwoDiamonds	800	2	2	100	0.941	0.779	
WingNut	1,016	2	2	100	1	1	
noisy_circles	1,500	2	2	100	1	1	
noisy_moons	2	2	100	1	1	
varied	2	3	95.87	0.96	0.89	
aniso	2	3	96.4	0.96	0.94	
blobs	2	3	100	1	1	
no_structure	2	1	100	1	1	
Revised DBSCAN Algorithm to Cluster Data with Dense Adjacent Clusters (Tran, T.N. et al)	Atom	800	3	2	98.875	0.988	0.9978	
Chainlink	1,000	3	2	100	1	1	
GolfBall	4,000	3	1	100	1	1	
Hepta	212	3	7	99.53	0.995	0.9942	
LSun	400	2	3	100	1	1	
Target	770	2	2	98.44	1	1	
Tetra	400	3	4	94	0.939	0.914	
TwoDiamonds	800	2	2	100	1	1	
WingNut	1,016	2	2	100	1	1	
noisy_circles	1,500	2	2	100	1	1	
noisy_moons	2	2	100	1	1	
varied	2	3	95.87	0.97	0.94	
aniso	2	3	99.07	0.99	0.98	
blobs	2	3	100	1	1	
no_structure	2	1	100	1	1	
Density-Connected Subspace Clustering for High-Dimensional Data (SUBCLU)	Atom	800	3	2	98.875	0.988	0.978	
Chainlink	1,000	3	3	96.8	0.6911	0.7207	
GolfBall	4,000	3	3	84.89	0.43	–	
Hepta	212	3	5	77.91	0.49	–	
LSun	400	2	3	1	1	1	
Target	770	2	2	98.44	1	1	
Tetra	400	3	4	58.25	0.59	0.36	
TwoDiamonds	800	2	2	100	0.941	0.779	
WingNut	1,016	2	2	100	1	1	
noisy_circles	1,500	2	2	100	1	1	
noisy_moons	2	2	100	1	1	
varied	2	3	95.87	0.96	0.89	
aniso	2	3	96.4	0.96	0.94	
blobs	2	3	100	1	1	
no_structure	2	1	100	1	1	
k-Means	Atom	800	3	2	100	0.29	0.18	
Chainlink	1,000	3	2	0.493	–	
GolfBall	4,000	3	1	1	1	
Hepta	212	3	7	1	1	
LSun	400	2	3	0.28	0.24	
Target	770	2	2	0.266	0.2	
Tetra	400	3	4	1	1	
TwoDiamonds	800	2	2	1	1	
WingNut	1,016	2	2	0.89	0.67	
noisy_circles	1,500	2	2	100	0.49	–	
noisy_moons	2	2	0.85	0.49	
varied	2	3	–	0.81	
aniso	2	3	0.83	0.61	
blobs	2	3	1	1	
no_structure	2	1	1	1	

Time performance tests

In our time performance analysis, we compare the time spent in seconds for the several clustering algorithms to complete execution for each dataset run. We use the original DBSCAN algorithm’s running time as the base value in to understand how our proposed algorithm scales compared to the other target algorithms.

The traditional DBSCAN algorithm exhibits a time complexity where the RegionQuery module is O(n), leading to an overall complexity of O(n2) (Ester et al., 1996; Bushra & Yi, 2021). However, the use of structures like the R*-tree can reduce this to O(nlogn). The proposed algorithm merely introduces an additional condition for searching the ClosestCore within the original DBSCAN ( Algorithm 1 ), thus not altering the complexity which remains at O(n2) or O(nlogn). However, due to the inclusion of the ClosestCore search condition, experimental results indicate a marginal linear increase in time compared to DBSCAN. Tran, Drab & Daszykowski (2013) is O(n2) in the worst case, similar to DBSCAN, but performs additional computations to separate clusters of dense regions. OPTICS is also O(n2) in the worst case, but performs additional operations to align the objects (Ankerst et al., 1999). These additional computational tasks mean that both algorithms require more computational resources than DBSCAN, and as the data size grows, experimental results show a linear increase in time compared to DBSCAN. Finally, for SUBCLU, it operates clustering within subspaces, and the number of subspaces can increase exponentially with the number of dimensions (Kailing, Kriegel & Kröger, 2004). Consequently, in the worst-case scenario, it presents a complexity of O(2dn2). It is observable that with the increase in the number of data points and dimensions, the time taken by this algorithm is longer in comparison to other algorithms. The time performance results are reported on Table 9. Figure 5 shows a graphical representation of the runtime of the algorithms for a better visualization of how each algorithm scales up together with the size of the geo-tagged twitter dataset instances.

Table 9 Time performance results for the twitter dataset instances and the real-world multi-dimensional dataset collection.

Datasets	Algorithms	
	DBSCAN	AutoSCAN	Tran, T. N. et al	OPTICS	SUBCLU	
Name	Size	Feature	Time elapsed (s)	Time elapsed (s)	Time elapsed (s)	Time elapsed (s)	Time elapsed (s)	
Geo-Tagged Twitter Dataset	1000	2	0.063914	0.83378	0.077539	0.7465	0.492	
10,000	3.48978	4.93034	4.3934	11.37043	103.11	
50,000	87.467	233.714	109.711	163.2778	3002.34	
100,000	353.202	372.509	462.188	517.211	24672.01	
1000,000	34847.4	39113.74	51018.3	61955.6	349034	
2000,000	138276	148543	199842	254701	–	
Iris	150	4	0.019372	0.027221	0.026467	0.07911	0.10344	
TAE	980	10	0.019735	0.02633451	0.020618	0.07789	0.93302	
Wine	178	13	0.0319	0.041023	0.0405	0.08694	0.31046	
MFCC	7,195	22	39.4394	11.9374186	57.7131	62.10119	175.0385	
Ecoli	336	8	0.06271	0.0742318	0.079497	0.14022	0.302291	
Yeast	1,484	8	1.13692	1.1818285	1.4967	1.5002	2.129	
Glass	214	10	0.04331	0.046507	0.047	0.10462	0.3991	
WDBC	569	30	0.29935	0.4260416	0.31669	0.3925	5.3945	

Figure 5 Time performance comparisons.

Time performance comparisons of our proposed algorithm, AutoSCAN, the DBSCAN, Tran, Drab & Daszykowski (2013) algorithm, and the OPTICS algortihm. As the instance of the geo-tagged twitter dataset grows, AutoSCAN the least increase from the original DBSCAN implementation.

From the geo-tagged twitter instance dataset results, the experimental results show that AutoSCAN performs faster than other previously proposed clustering algorithms. Furthermore, we can see how linearly our method scales as the instances of the twitter datasets grow logarithmically when compare to Tran, Drab & Daszykowski (2013) and the OPTICS approach. Even though the AutoSCAN does run slightly slower than the original DBSCAN algorithm, we have shown that the increase in clustering quality makes up for this additional runtime. Meanwhile, the results collected from running the multi-dimensional datasets suggest our algorithm’s complexity is not significantly affected by the feature size of a dataspace. The same cannot be said for the subspace clustering algorithm SUBCLU as it shows a considerable increase in runtime in cases like the “MFCC” dataset that exhibit 22 feature size in the dataspace.

In our experiment section, we were able to validate our two proposed methods by running accuracy tests on multiple benchmark datasets, each with their own specific challenges, designed to test the overall capacity of a clustering algorithm. We show that our first proposed algorithm can accurately determine the ɛ parameter of the DBSCAN algorithm in a range that yields optimal performance. We also demonstrated that our second proposed algorithm improves the clustering accuracy by identifying appropriate cluster labels through additional operations on boundary objects that appear in overlapping density regions. In the comparative analysis section, six previously proposed density-based clustering algorithms including the original DBSCAN were used to compare the accuracy and time performance results. From these tests, the AutoSCAN was able to identify the best clustering quality on the FCPS and scikit-learn benchmark datasets. The time performance results show that while AutoSCAN scales faster than the previously proposed density-based clustering algorithms, it computes at a slightly slower rate than the original DBSCAN. However, this increase in computation of AutoSCAN compared to DBSCAN is made up by its better accuracy.

Conclusion

In this article, we propose an unsupervised method to determine the optimal DBSCAN parameters from its density distribution. We show that by extracting a dataset’s k-nearest neighbor information, and calculate its population distribution, we were able to identify the appropriate cut-off region that best fits the definition of a cluster in terms of the DBSCAN algorithm. We employ a cubic B-spline curve-fitting function on the density distribution to iterate through the possible ɛ values and generate the optimal value. Additionally, an efficient method to cluster data in overlapping density regions is discussed that uses the implementation from the original ExpandCluster module and identifies the correct border objects for each groups of data without additional complexity. Finally, the experimental analysis demonstrates that AutoSCAN can determine the optimal parameter ɛ and that it improves the clustering quality compared to that of the original DBSCAN algorithm through the accurate clustering of border objects. It also shows a higher accuracy and lower running time than other density-based clustering algorithms.

Supplemental Information

Supplemental Information 1 Codes for automatic detection of DBSCAN parameters and efficient clustering of data in overlapping density regions

Additional Information and Declarations

Competing Interests

Author Contributions

Data Availability

The authors declare there are no competing interests.

Adil Abdu Bushra conceived and designed the experiments, performed the experiments, analyzed the data, performed the computation work, prepared figures and/or tables, authored or reviewed drafts of the article, and approved the final draft.

Dongyeon Kim performed the experiments, analyzed the data, prepared figures and/or tables, authored or reviewed drafts of the article, and approved the final draft.

Yejin Kan conceived and designed the experiments, prepared figures and/or tables, and approved the final draft.

Gangman Yi conceived and designed the experiments, analyzed the data, authored or reviewed drafts of the article, advising and fund, and approved the final draft.

The following information was supplied regarding data availability:

The source code is available in the Supplemental File.

The data is available as follows:

- Clustering and Projection Suite (FCPS) Thrun, M. C. and Ultsch, A. (2020). Clustering benchmark datasets exploiting the fundamental clustering problems. Data in Brief, page 105501

https://data.mendeley.com/datasets/vsxvgc4rwy/1

- The scikit-learn synthetic datasets by Pedregosa, F., Varoquaux, G., Gramfort, A., Michel, V., Thirion, B., Grisel, O., Blondel, M., Prettenhofer, P., Weiss, R., Dubourg, V., et al. (2011). Scikit-learn: Machine learning in python. Journal of machine learning research, 12(Oct):2825–2830.

https://scikit-learn.org/stable/auto_examples/cluster/plot_cluster_comparison.html

- The National Center for Supercomputing Application (NCSA) collected the real-world dataset at https://b2share.eudat.eu/records/7f0c22ba9a5a44ca83cdf4fb304ce44e

- UCI Machine Learning Repository: http://archive.ics.uci.edu/ml.

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
