# Peer review of "AutoSCAN: automatic detection of DBSCAN parameters and efficient clustering of data in overlapping density regions"

_PeerJ Computer Science, doi:10.7717/peerj-cs.1921_

## Round 0.1 · original submission · Major Revisions

Please consider the reviewers comments.

·

Basic reporting

### English language

Clear, correct and on professional level.

### Intro, background and references

The abstract and introduction clearly introduce the reader to the problem stated in the text. The introduction contains a literature survey that is relevant to presented research and clearly shows where is the place of the described method among the existing clustering approaches.

### Text structure

The structure of the text is very good and it fulfills the requirements of the journal. Obviously authors used LaTeX for manuscript preparation and the technical quality of the presented text is very high, including the mathematical expressions and the pseudocode of the presented algorithm.

### Figures and tables

Tables are well structured and are clear. The font in Table 1 looks too small, maybe there can be found a solution to make it bigger and easier to read.

Figures are also very well structured, and they clearly represent the idea behind them. However in this format the text included in the figures becomes too small and hard to read. I suggest the authors either to restructure the figures, or maybe to to rotate them, so the font of the text will become relatively bigger. (See Fig. 2, 3, 4). In Fig. 5, 6 authors can directly make the font larger.

### Raw data

Authors share the source code which consists of a program in C++ and a Python script. The code is written in high professional level, and it is well documented.

Experimental design

### Originality of the research and scope of the journal

The presented clustering method is original and it is within the scope of the journal.

### Research questions definition

The topic of the research is clear and well defined. The authors give clear arguments why their modification is needed.

### Technical and ethical standards of the research

The mathematical background of the presented approach and the algorithm of the solution are clearly stated.

### Description detail sufficiency to replicate

The description is clear enough, so the experiments can be replicated.

Validity of the findings

### Impact and novelty

Clustering algorithms are an important topic in computing. The presented modification/algorithm is novel.

### Data provided robustness and statistical control

The experimental verification of the complexity of the algorithm is not enough. This is the only drawback in this paper proposition. Please, provide analysis and proof of the computational complexity of your algorithm in the terms of asymptotic notation. Provide a mathematical proof of the complexity of your algorithm.

### Conclusions

Conclusions are clear and provide a clear summary of the results.

Additional comments

Please, fix the figures a bit, the font in the table, provide proof for computational complexity of your algorithm. The text is a very good proposition for a journal paper.

Reviewer 2 ·

Basic reporting

The paper proposed a density-based clustering approach to address dependency of DBSCAN on the clustering parameters and deficiency in handling overlapping clusters. The problem is worthy of investigation and details of the proposed solutions are presented. Several concerns or suggestions are listed below:
1. The contribution section needs to further revised. As I understood from the authors, two major contributions would be the selection of clustering parameters and the enhancement of clustering highly overlapped clusters. The remaining contributions are more like experimental analysis and comparison studies, which can be combined.
2. For the Background section, the authors should just provide the preliminary definitions or introductions about the DBSCAN algorithm. For the related work section, I would suggest the authors to make it a separate section and move it before the Background Section. Also, the authors should provide a more detailed survey on different clustering methods, including partition-based, grid-based, density-peak clustering, etc..
3. In the Proposed Methodology section, are there any explanations or justification on the specification of k? How does this k value affect the overall clustering performance?
4. For the automatic detection of Epsilon Parameters, the authors proposed an effective strategy that is similar to the idea used in density-peak clustering approaches. I would suggest the authors to emphasize their difference with reference to the following works:

** Staff Note: For the following citation suggestions, it is PeerJ policy that additional references suggested during the peer-review process should only be included if the authors are in agreement that they are relevant and useful **

• Rodriguez, A., & Laio, A. (2014). Clustering by fast search and find of density peaks. science, 344(6191), 1492-1496.
• Chen, Y., Hu, X., Fan, W., Shen, L., Zhang, Z., Liu, X., ... & Li, H. (2020). Fast density peak clustering for large scale data based on kNN. Knowledge-Based Systems, 187, 104824.
• Hou, J., Zhang, A., & Qi, N. (2020). Density peak clustering based on relative density relationship. Pattern Recognition, 108, 107554.
• Yan, X., Homaifar, A., Nazmi, S., & Razeghi-Jahromi, M. (2017, October). A novel clustering algorithm based on fitness proportionate sharing. In 2017 IEEE International Conference on Systems, Man, and Cybernetics (SMC) (pp. 1960-1965). IEEE.
5. For the Efficient Clustering in Overlapped regions, more explanations are needed to clarify the difference between closestCore and closestcore. Do they have the same meaning?

Experimental design

Sufficient details about the experiments are provided and extensive experiments are presented to support the efficacy of the proposed approach using a variety of benchmark datasets.

Validity of the findings

Impact and novelty of the proposed approach are well justified in the experiments. However, more detailed explanations about the proposed approach as well as more related works should be included especially for density-peak clustering.

Additional comments

NA.

---

## Round 0.2 · accepted · Accept

Based on the reviewers, the paper can be accepted.

·

Basic reporting

The authors have taken into account my remarks from the previous review. My opinion is that this text is a very good proposition for a journal paper in the field of computer science.

Experimental design

n/a

Validity of the findings

n/a

Additional comments

n/a